# Systematic Inflammation and Oxidative Stress Elevation in Diabetic Retinopathy and Diabetic Patients with Macular Edema

**DOI:** 10.3390/ijms26083810

**Published:** 2025-04-17

**Authors:** Kamelia Petkova-Parlapanska, Valeria Draganova, Ekaterina Georgieva, Petya Goycheva, Galina Nikolova, Yanka Karamalakova

**Affiliations:** 1Medical Chemistry and Biochemistry Department, Medical Faculty, Trakia University, 11 Armeiska Str., 6000 Stara Zagora, Bulgaria; kamelia.parlapanska@trakia-uni.bg (K.P.-P.); yanka.karamalakova@trakia-uni.bg (Y.K.); 2Department of Otorhinolaryngology and Ophthalmology, Medical Faculty, Trakia University, 6000 Stara Zagora, Bulgaria; valeriya.draganova@trakia-uni.bg; 3Department of General and Clinical Pathology, Forensic Medicine, Deontology and Dermatovenerology, Medical Faculty, Trakia University, 11 Armeiska Str., 6000 Stara Zagora, Bulgaria; ekaterina.georgieva@trakia-uni.bg; 4Propaedeutic of Internal Diseases Department, Medical Faculty, Trakia University Hospital, 6000 Stara Zagora, Bulgaria

**Keywords:** oxidative stress, diabetic retinopathy, inflammation markers

## Abstract

This study investigates the association between diabetic retinopathy (DR) and its complication, diabetic macular edema (DME), and compared it with biomarkers of oxidative stress. This study aimed to compare the main indicators of the development of diabetic retinopathy measured as parameters of oxidative stress and compared to lipid oxidation, DNA damage, and cytokine levels and to monitor their quantitative manifestation in DME. This study evaluated 134 patients (62 males and 72 females; aged 62.10 ± 11.22 years) and divided them into two groups: type 2 diabetes mellitus with DR and type 2 diabetes mellitus with DME. All results were compared with healthy volunteers (*n* = 94) and showed that patients with DME had significantly higher levels of ROS, cytokine production, lipid oxidation, and DNA damage. In addition, patients with DME had decreased levels of nitric oxide (NO) and an impaired NO synthase (NOS) system (*p* < 0.05). These findings suggest that patients with DR and DME are unable to compensate for high levels of oxidative stress. Reduced NO levels in patients with DME may be due to impaired NO availability. This study highlights compromised oxidative status as a contributing factor to DME in patients with decompensated type 2 diabetes mellitus. An assessment of oxidative stress levels and inflammatory biomarkers may aid in the early detection and prediction of diabetic complications.

## 1. Introduction

Diabetic retinopathy (DR) is a complication associated with diabetes that primarily arises from damage to the small blood vessels in the light-sensitive part of the retina due to prolonged high blood sugar levels. Diabetic macular edema (DME) is the most severe complication of DR. It is characterized by fluid leakage and the formation of new abnormal blood vessels. Macular edema can lead to reduced visual acuity and, if it becomes chronic and refractory, can result in severe and permanent vision loss or blindness [1]. DME is one of the leading causes of blindness among working-age individuals in developed countries, highlighting its significant medical and socio-economic implications [2]. Chronic hyperglycemia is a major factor contributing to retinopathy. It leads to an increased formation of advanced glycation end products (AGEs) and oxidative stress (OS), which in turn changes the structure and function of blood vessels [3]. Damaged mitochondria are known to contribute to the production of reactive oxygen species (ROS) in cells. In diabetes, the retina faces a dual challenge, as both the enzymatic and non-enzymatic defense systems become compromised. Retinal ischemia leads to the production of vascular growth factors, which can increase the permeability of the blood–retinal barrier. This results in fluid leakage into the retinal tissue, causing edema [4]. Additionally, diabetes elevates the levels of various inflammatory molecules, including cytokines and chemokines, within the retina. Patients with diabetes who have proliferative retinopathy show heightened levels of inflammatory mediators, such as tumor necrosis factor-alpha (TNF-α), interleukin-1 beta (IL-1β), soluble interleukin-2 receptor (sIL-2R), and interleukin-8 (IL-8) [5]. The increased oxidative stress damages retinal endothelial cells, resulting in greater microvascular permeability and promoting the recruitment of new inflammatory cells [6]. For instance, interleukin-17A (IL-17A) plays a role in the early stages of diabetic retinopathy development [7,8]. Together, oxidative stress, inflammation, angiogenesis, and apoptosis contribute to the pathogenesis of DR and DME, either independently or collectively [9,10,11,12]. These processes lead to functional damage to vascular endothelial cells; alterations in retinal neurons; and ultimately, the apoptotic death of these cells, which are key features of DR and DME [13,14,15,16].

This study aims to compare the key indicators of diabetic retinopathy development, focusing on three primary aspects: (1) the levels of ROS and RNS as measures of oxidative stress, (2) lipid oxidation and DNA damage, and (3) cytokine production. Additionally, we will examine the quantitative manifestations of these indicators in DME. Identifying fundamental biomarkers for routine assessment in significant diseases holds great potential for predicting and preventing subsequent complications. This study highlights electron paramagnetic resonance (EPR) spectroscopy as a promising method for the direct measurement of NO in clinical settings, potentially serving as a routine blood test to predict complications associated with diabetes mellitus.

## 2. Results

### 2.1. Main Biochemical Serum Parameters in the Studied Patients

Table 1 shows that the activity of antioxidant enzymes—specifically superoxide dismutase (SOD; *p* < 0.001); catalase (CAT, *p* < 0.001); and glutathione peroxidase (GPx, *p* < 0.001)—is significantly lower in diabetes mellitus groups with complications compared to the control group. Furthermore, within the complicated groups, there is a notable difference in enzyme activity; the DME group exhibits lower activity than the DR group (*p* = 0.00).

Table 2 shows the values of central retinal thickness in the macular region in patients with DME, with DR without DME, and a healthy control group. The results prove the statement that the values from clinical studies exceed 300 µm in DME and do not reach 230 µm in patients with DR without DME. It is clear from the data in the table that the control group is healthy individuals without manifest macular edema. According to the American Academy of Ophthalmology (AAO), diabetic macular edema is a thickening of the retina in the macula region, caused by the accumulation of fluid in the intraretinal space in the area of the inner and outer plexiform layers. It is due to hyperpermeability of the retinal vessels and may be present at all stages of development of diabetic retinopathy. Its pathophysiological mechanism involves passive leakage of fluid due to high hydrostatic pressure exerted on the retinal capillaries. The integrity of the inner and outer blood–retinal barriers is compromised, resulting in a disruption of the vitreomacular interface.

Diabetic macular edema (DME) is diagnosed using a combination of quantitative imaging and clinical examination. Multiple studies have reported that optical coherence tomography (OCT) is the primary measurement tool, with central retinal thickness thresholds most commonly defined as between 230 and 300 μm, while other studies have defined DME as a central macular thickness exceeding 290 μm. In parallel, clinical methods based on the ETDRS criteria identify DME by the presence of retinal thickening—especially when it involves or threatens the center of the macula—as observed by fundus photography or biomicroscopy. According to the ETDRS, the criteria for defining DME as clinically significant are as follows:-Thickening approximately 500 μm from the center of the macula.-Hard exudates around the center of the macula, in case the underlying retina is also thickened.-Presence of an area of retinal thickening with a diameter of about 1500 µm or more, located at a distance of 1DD or less from the center of the macula.

### 2.2. Influence of the Duration of Type 2 Diabetes on the Development of Diabetic Retinopathy

Diabetic retinopathy is underrepresented among diabetics with disease duration ≤ 10 years (7.5%; Figure 1). The incidence of this vascular-related manifestation in patients with a clinical course of >10 years of disease increases to 39.6%, which is more than a 5-fold increase (*p* < 0.001) (Figure 1). Therefore, disease duration is essential for the possibility of developing retinopathy.

### 2.3. Influence of Disease Duration on the Manifestation of DME

Of the studied group of diabetics with retinopathy (96 patients), 38 had DME—a relative share of 39.5%.

The manifestation of DME is observed about 15 years from the onset of the disease, and the average age is 63 years (Figure 2). A multiplied mode is established—35, 52, and 60—which occurs 2 times. The median is equal to 63 years. No gender difference was found to influence the manifestation of the complication.

### 2.4. Levels of ROS and AGEs

Measuring ROS levels is a suitable method to determine the levels of oxidative stress. ROS possesses a direct relationship with AGE/RAGE signaling, inflammation, and endothelial function [17]. The reported results show that there was no significant difference in the levels of AGEs between the DR group and the DME (13.94 ± 0.31 vs. 14.28 ± 0.31). In both studied groups with DMT2, extremely high levels of AGEs were observed compared to the control group (13.94 ± 0.31 vs. 14.28 ± 0.31 vs. 2.73 ± 0.21) (Figure 3).

### 2.5. Serum Levels of NO, Endothelial Nitric Oxide Synthase (eNOS), and Inducible iNOS

Enzymatic pathways that can generate high glucose-induced ROS were investigated. Nitric oxide levels were chosen, because it is a gaseous lipophilic free radical that plays an important role in vascular physiology and is known as endothelial relaxing factor [18].

The DMΕ group showed significantly lower NO• levels compared to the control (*p* = 0.00) (Figure 4A), as did the DR group (*p* = 0.00). The concentrations of eNOS (mean 69.04 ± 1.21; Figure 4B) compare to iNOS (mean 74.66 ± 1.68; Figure 4C) in the DME group were lowest. There was also a statistically significant difference in the DR groups’ concentrations of eNOS (mean 457.60 ± 8.44 vs. mean 305.18 ± 5.85; *t*-test) and iNOS (mean 249.38 ± 2.96 vs. mean 162.14 ± 1.86; *t*-test) vs. control.

### 2.6. Mean Serum Levels of the Pro-Oxidant Malondialdehyde (MDA) and Pro-Oxidant Molecules 4-Hydroxy-2-Nonenal (4-HNE)

Lipid peroxidation is commonly used as a marker of oxidative damage to lipids. In addition, lipid peroxides are also markers of initial free radical reactions and specific markers of cell membrane damage [19].

In DME (Figure 5A), MDA levels showed a statistically significant increase compared to controls (*p* = 0.00). In the DR group, MDA levels were also statistically significantly increased compared to the control group (DME, *p* = 0.00 and DR, *p* = 0.00). The results show a statistically significant difference in 4-HNE between the DR groups (*p* = 0.00) and DME (*p* = 0.000) and the control group (Figure 5B). The level of 4-HNE was significantly higher in DME patients (mean 45.94 ± 0.57) compared to controls (mean 15.01 ± 0.66) and the DR group (mean 31.55 ± 1.09).

### 2.7. DNA Oxidation Biomarker Study: 8-Hydroxy-2-Deoxyguanosine (8-OHdG) Level

8-hydroxy-2′-deoxyguanosine (8-OHdG), produced by ROS/RNS, is the most common marker of oxidative DNA damage. According to data, the determination of these markers would significantly contribute to a better understanding of the role of epigenetic mechanisms in the evolution of the complex disease diabetes [20,21,22,23].

Results for 8-OHdG (Figure 6) reveal significantly increased values of this biomarker in the group with DME (mean 6.25 ± 0.32) compared to the other two groups: control (mean 3.88 ± 0.19) and DR (mean 4.56 ± 0.27).

### 2.8. An Examination of the Levels of TNF-α, TGF-β, IFN-γ, IL-1β and IL-6, and IL-17A

Figure 7 shows the expression levels of inflammatory cytokines, including TNF-α, IL-1β and IL-6, and IL-17A, which attract monocytes and leukocytes and promote a continuous inflammatory response. These inflammatory processes lead to a decrease in local blood flow velocity, further increasing retinal hypoxia [24]. The levels of TGF-β and IFN-γ were significantly higher in the DME group compared to the control group (*p*  =  0.0001 and *p*  =  0.0001). Significantly higher values were obtained in the DME group and the DR group compared to the control group for interleukins IL-6 (mean 14.29 ± 0.49 vs. mean 12.68 ± 0.33 vs. mean 8.01 ± 0.21; *t*-test *p* < 0.05) and IL-17A (mean 112.09 ± 6.50 vs. mean 53,86 ± 1,88 vs. mean 7.12 ± 0.15; *t*-test *p* < 0.05). IL-1β levels were elevated in the control group, in contrast to the DME and DR groups (*p* = 0.0001 and *p* = 0.004). The DME group also had extremely higher serum levels of tumor necrosis factor-α (TNF-α) (mean 88.40 ± 0.70) compared to the DR and control groups (mean 59.02 ± 1.75 vs. mean 31.43 ± 0.99; *t*-test *p* < 0.05).

## 3. Discussion

Diabetic macular edema (DME) is a significant cause of blindness in working-age adults in developed nations, underscoring its substantial medical and socio-economic impact. Prolonged high blood sugar levels play a key role in the development of retinopathy by promoting the accumulation of advanced glycation end products (AGEs) and oxidative stress, which damage blood vessel structure and function [2,3]. Additionally, as illustrated in Figure 1, the duration of diabetes significantly influences the likelihood of retinopathy onset, while no gender-based differences have been observed in the occurrence of this complication (Figure 2).

The accumulation of AGEs, which form due to hyperglycemia, is a significant factor influencing the development of retinopathy. AGEs directly cross-link long-chain proteins, contributing to the progression of microvascular complications [25]. This explains the difference in AGE levels between healthy individuals and those with DR and DME (Figure 3). Hyperglycemia increases the generation of ROS. When glucose levels rise, glucose reacts non-enzymatically with proteins, leading to the formation of “Amadori products”, which subsequently form AGEs [18]. Additionally, ROS have vasoconstrictive effects by inhibiting the pathways responsible for endothelium-dependent vasodilation, such as those involving NO. The endothelium continuously produces NO from L-arginine through the action of eNOS. However, the primary mechanism for reducing vascular NO bioavailability is its rapid inactivation by ROS [26,27]. This study observed increased ROS levels in both participant groups—those with DR and those with DME. The analysis indicated that ROS is an independent risk factor for the development of DR.

A statistically significant difference was also observed between the two groups with complications (refer to Figure 4A). Increased ROS activate inflammatory factors, and in turn, the immune response elevates ROS formation in a stepwise manner, which may explain the heightened levels seen in the DME group [28,29]. Furthermore, there is evidence that persistent oxidative stress causes eNOS to become dysfunctional, resulting in the production of superoxide anion instead of NO. The biological roles of eNOS in the pathogenesis of DME include the following: (1) the induction and retention of inflammatory cells in the ocular microcirculation; (2) a direct effect on cell junction proteins, which reduces their expression; and (3) an increased expression of the vascular permeability factor, vascular endothelial growth factor A [30]. The findings of this study confirm that NO radicals are crucial in the oxidative pathways associated with diabetic complications, such as DR and DME, playing a central role in vascular damage. One significant result of the analysis indicates remarkably reduced serum levels of NO, with notably lower values among diabetics with vascular complications. All patients in this group exhibited extremely low NO levels compared to healthy individuals. Thus, this marker can serve as an indicator for both complications (see Figure 4). The activity of antioxidant enzymes, such as SOD, CAT, and GSH-Px, is significantly reduced in groups examined with complications. In contrast, healthy individuals, including those who are obese, maintain normal levels of these antioxidant enzymes [31]. Notably, SOD is overexpressed in adipose tissue in cases of obesity, which helps prevent the development of fatty liver and insulin resistance. However, lower levels of SOD and CAT have been observed in diabetic patients compared to healthy controls. Additionally, there are significant variations in the concentrations of these enzymes relative to the severity of DR. It has been reported that SOD and CAT levels are considerably reduced in patients with DR compared to those with diabetes who do not have ocular complications [32].

Karan et al. [33] stated in their study that specific lipid aldehydes, including 4-HNE, and MDA are strongly associated with the onset and progression of DR, a finding that aligns with our clinical study. We observed that plasma concentrations of 4-HNE in patients with DME are higher than those in individuals with diabetic retinopathy. Additionally, levels of 4-HNE show a positive correlation with plasma levels of MDA (see Figure 5). The analysis of oxidant profiles among the diabetic groups revealed significantly elevated MDA levels (Figure 5). The cohort of patients with complications from DMT2 shows that mean serum levels of MDA were notably higher compared to healthy individuals. In addition, prooxidant molecule levels were found to be significantly elevated—five times higher—in patients with DME complications. Other studies have indicated that MDA and 4HNE may serve as potential biomarkers for assessing the risk of DR and DME in patients with DMT2 [34]. Other researchers, conducting experiments on vascular smooth muscle cells, discovered that the expression of lipoxygenase induced by 4-HNE is closely linked to NO production [35]. They found that NO present in ocular tissues plays a crucial role in the progression of diabetic eye damage [36]. Based on presented results, we concluded that 4-HNE is also significant in the development of DME. The accumulation of DNA damage products, such as 8-OHdG, along with LPO end products, has been linked to age-related macular degeneration. RNS, particularly peroxynitrite (ONOO−), react with guanine and lead to both oxidative and nitrative DNA damage, resulting in compounds like 8-oxodeoxyguanosine and 8-nitroguanine [37]. This suggests that the systemic production of 8-OHdG is associated with dysregulated biochemical processes, which can culminate in secondary damaging effects on small blood vessels [38]. When evaluating the role of 8-OHdG in the development of DR and DME, elevated levels of this biomarker have been reported in patients with DME [39,40]. Research indicates that retinal cells exposed to high levels of glucose, superoxide, and hydrogen peroxide are particularly vulnerable to ROS-induced damage [41,42]. This damage includes both membrane lipid peroxidation and oxidative DNA damage, as measured by 8-OHdG, which has been observed in mouse models [43,44].

Interleukin 17 (IL-17A) plays an important role in a wide variety of immunological diseases, but at the ocular surface, it promotes neutrophil infiltration into tissues that induce the synthesis and secretion of matrix metalloproteinases and ROS, which generate disruption of the corneal epithelium, loss of epithelial functionality, and induction of apoptosis [45]. In terms of inflammatory processes that promote the development of diabetic retinopathy, measurements have shown exceptionally high levels of IL-17A in patients with DME. Zho et al. [46] explained that hyperglycemia activates the Retinoid-Related Orphan Receptor (RORγt) in circulating Th17 (Th17) cells, which induces the production of IL-17A. Some of these circulating Th17 cells adhere to the retinal vasculature, disrupt the blood–retinal barrier, and release IL-17A into the retina, which binds to the IL-17A receptor (IL-17R) on Müller glia and photoreceptors. Once IL-17A binds to IL-17R, an adaptor molecule known as Act1 activates Nuclear factor kappa B NF-κB, which induces endothelial cell death. This leads to retinal vascular damage and the development of non-proliferative diabetic retinopathy [47,48,49,50,51,52]. Since ROS are crucial for the activation of proinflammatory mediators, such as NF-kB [53], this is likely to account for the positive association between ROS and IL-17A. Other interleukins, IL-6 and TNF-α, have been identified as major predictors of DME [54]. When analyzing the experimental results in this study, high levels of TNF-α and IL-6 were observed (Figure 7). In patients with DME, an increase in TNF-α value of about 30% was found. Increased levels of TNF-α have also been found in plasma, serum, vitreous humor, and platelets of patients with diabetes [55], and these increases in plasma and serum are associated with “worse” DR. It has also been shown that TNF-α enhances angiogenesis and is required for VEGF-induced endothelial hyperpermeability [56]. The results of the performed correlation analysis showed a positive correlation between IL-17A and TNF-α, as well as between TNF-α and IFN-γ. Previous studies have commented on the synergism of IL-17A with strong NF-kB activators, such as TNF-α, IL-1β, and IFN-γ [57]. The molecular mechanisms by which many of these synergistic events act on IL-17 signaling are not yet elucidated [58].

As a key player in inflammation, IFN-γ can be found in various parts of the eye and is responsible for the breakdown of the blood–retinal barrier and the activation of inflammatory cells and other cytokines that accelerate neovascularization and neuroglial degeneration. In addition, IFN-γ is involved in other vascular complications of diabetes mellitus, such as diabetic nephropathy, cerebral microbleeds, and age-related macular degeneration [59,60]. Of particular interest are the measured mean serum levels of IFN-γ, which were approximately 15% higher in the RD group compared to controls and 46% higher in the DEM group compared to RD. Evidence also describes the significant role that TGF-β signaling plays in the progression of age-related macular degeneration [61]. The median age of onset of DME, as determined by us, was 63 years. Most likely, both age and diabetes determine the high values obtained for TGF-β in the DME group (Figure 7).

The findings regarding IL-1β levels are important (see Figure 7). There were no significant differences in IL-1β levels among the groups with DME, DR, and the control group. While elevated IL-1β levels have been noted in the retinas of diabetic mice [62,63,64], the levels in healthy adults are still unclear. IL-1β is linked to diabetic nephropathy, cardiovascular complications, and conditions like polymyalgia. High glucose levels in diabetes promote macrophage polarization, leading to increased IL-1β release. Moreover, IL-1β and VEGF mutually regulate each other in endothelial cells, both playing vital roles in angiogenesis [65]. Yoshida et al. [66] highlighted IL-1β as an important marker for treating DME and preventing early DR. In contrast, Li et al. [67] found no significant difference in IL-1β levels between the DME and control groups, matching the results of the present study (see Figure 7C). More research with humans on this interleukin is needed.

Additionally, Andrés-Blasco et al. [68] identified MDA, VEGF, and IL-6 as contributors to DR and DME. These findings help in identifying biomarkers for patients with DMT2 at risk of DR, DME, and vision loss. The presence of cases resistant to anti-VEGF treatment indicates that other factors may also play a role in these conditions [69,70].

### Limitations

Oxidative stress and inflammation are interrelated processes and accompany many medical conditions, with the most common general redox and immune imbalance being an overlay between all the present diseases. Increased production of prooxidants by the mitochondrial electron transport system, oxidases, oxygenases, and nitric oxide synthase, together with reduced or ineffective detoxification, lead to oxidative stress in patients with chronic diseases and those with chronic medical conditions and DR-DME and ultimately irreversible damage to. In DR-DME patients, along with impaired redox homeostasis, high levels of inflammatory cytokines, adipokines, mitochondrial dysfunction, and low levels of endogenous antioxidants are reported, which leads to a chronic inflammatory state. Measurement of markers of inflammation and oxidative damage, together with general oxidative stress, will present us with the state of the organism at the time of the study. A major limitation of the methods and the combined analysis of total oxidative stress and markers of inflammation and ROS-induced damage to proteins, lipids, and nucleic acids is the differentiation of these parameters in DR and DME from those due to DR-DME in combination with other chronic diseases, such as hypertension, viral diseases, etc., which were not reported by the patients.

-Another limitation is the need to study several redox and inflammatory markers, since monitoring a limited number of them could not provide a complete picture of the redox state of the body. This requires the introduction of a multidisciplinary approach and personalized medical assessment.-At the same time, not all laboratories have the full range of specific equipment for the analysis, for example, an EPR spectrometer, which is the gold standard in assessing redox imbalance and levels of oxidants in the body. Instead, less sensitive and less specific spectrophotometric methods are used, which produce a high level of error. The assessment of redox imbalance and ROS and RNS levels by EPR introduces the need to create a single protocol for the analysis of serum samples from patients with DR-DME, which is currently not registered and unified.-There is individual variability between individual patients in terms of metabolism, body mass index (BMI); genetic and epigenetic factors; the activity of single-gene antioxidant systems (SOD, GPx, etc.); intracellular ROS metabolism (including mitochondrial damage); immune response (hyperinflammatory phenotype or immune dysfunction); presence of dysbiosis, etc. In certain patients, a non-specific correlation between individual markers can be observed, which explains why some patients with diabetes develop severe DR-DME compared to others despite similar therapeutic regimens.-Diet and intake of nutritional supplements with antioxidant properties, such as alpha-lipoic acid, vitamins, glutathione, or medications, for the management of DR-DME and chronic diseases that can influence or modify oxidative stress and inflammation to one degree or another are essential.

## 4. Materials and Methods

### 4.1. Ethics Statement

This work was conducted according to the Declaration of Helsinki and approved by the Ethics Board, Clinic of “Clinic for Endocrinology and Metabolic Diseases”, UMHAT “Prof. St. Kirkovich”, in Stara Zagora, Bulgaria. Written informed consent (2021/2023 MF, TrU, Stara Zagora) was obtained from the patients after hospitalization between January 2023 and August 2024.

### 4.2. Ophthalmologic Examination

Individuals with diabetes were classified as having fasting plasma glucose levels > 126 mg/dL or random blood glucose >200 mg/dL and a previous diagnosis of diabetes by an internist. Inclusion criteria included patients who had regular medical check-ups and required blood tests (HbA1c and creatinine) and urinalysis (urine protein and urine albumin). Patients with HbA1c data at least once every 6 months and for at least the past 7 years were included (Table 1). Exclusion criteria were lack of HbA1c data for the past 7 years and inability to perform urinalysis due to chronic renal failure. Patients were tested for visual acuity, and visual acuity was assessed using standardized charts (Snellen, LogMAR) to determine the level of central vision loss. Fundoscopy after mydriasis was performed—examination with direct/indirect ophthalmoscopy or slit lamp to detect the following signs: retinal thickening, hard exudates in the macular area, microaneurysms, hemorrhages, and cotton wool spots. The final diagnosis was confirmed using optical coherence tomography, Heidelberg Spectralis, OFTAS, Policoro, Italy.

The venous blood of fasting patients and healthy volunteers was collected in the morning after overnight fasting for lipid profile analysis. All collected samples for the electron paramagnetic resonance (EPR) study were studied immediately and then frozen at −80 °C for the ELISA test. Fasting plasma glucose (FPG) concentrations and glycated hemoglobin (HbA1c%), representing an average measure of glycemic exposure over time, were measured as parameters for glycemic control. The experiment was conducted following the ethical standards of the Medical Faculty and Hospital Research Committee and the Helsinki Declaration of 1964 and its later changes or comparable ethical standards. The estimated glomerular filtration rate (eGFR), in each case calculated from the level of creatinine in the blood and age, was calculated. After analysis of the data, the division of the patients into two groups of type 2 diabetes mellitus with DR and type 2 diabetes mellitus with DME was made. The new Fukuda classification was used to establish diabetic retinopathy [71].

### 4.3. Electron Paramagnetic Resonance (EPR) Study

All Electron Paramagnetic Resonance (EPR) measurements of all tested samples were conducted at room temperature (18–23 °C) on an X-band EMXmicro, spectrometer Bruker, Bremen, Germany, equipped with standard Resonator. Quartz capillaries were used as sample tubes. The sample tube was sealed and placed in a standard EPR quartz tube (i.d. 3 mm) which was fixed in the EPR cavity. All EPR experiments were carried out in triplicate and repeated. Spectral processing was performed using Bruker WIN-EPR and SimFonia software 2021.

#### 4.3.1. An Evaluation of the ROS Product Levels

ROS levels were determined according to the method of Shi et al. [72] with some modifications. Ex vivo EPR spectroscopy was used to study ROS formation in real time in the serum of patients and controls. *N*-tert-butyl-alpha-phenylnitrone (PBN) was used as a spin-trapping agent. PBN, upon reaction with unstable radicals, such as ROS, forms relatively stable spin adducts that can be subsequently detected by EPR spectroscopy. Briefly, 900 μL of 50 mM PBN dissolved in dimethyl sulfoxide (DMSO) was added to 100 μL of plasma, and after centrifugation at 4000× *g* rpm for 10 min at 4 °C, EPR spectra were immediately recorded in the supernatant. The levels of ROS products were calculated as double-integrated plots of the EPR spectra, and the results were expressed in arbitrary units (a.u.). The EPR settings were as follows: 3503.73 G, center field; 20.00 mW, microwave power; 5 G, modulation amplitude; 50 G, scan width; 1 × 10^5^, gain; 81.92 ms, time constant; 125.95 s, scan time; 5 scans per sample.

#### 4.3.2. An Evaluation of the •NO Radical Levels

Laboratory measurement of NO radicals is extremely difficult due to their biochemical instability, short half-life, and sometimes very low concentration in biological fluids. EPR offers very high accuracy, using a spin trap, such as Carboxy-PTIO, by forming a stable adduct with the radical that exists long enough to be measured. To a 50 μM solution of carboxy 2-(4-carboxyphenyl)-4,4,5,5-tetramethyl is added the potassium salt of imidazoline-1-oxyl-3-oxide (carboxy PTIO.K), dissolved in a mixture of 50 mM Tris (pH 7.5) and DMSO in a ratio of 9:1.

### 4.4. Enzyme-Linked Immunosorbent Assay

The markers of oxidative stress were measured with ELISA kits following the manufacturer’s instructions. The ELISA kits were as follows: Human AGE levels ELISA Kit (ab238539); Human Malondialdehyde (MDA) (ab233471), 4-hydroxy-2-nonenal 4-HNE, Human eNOS (ab241149); Human iNOS ELISA Kit (ab253217); Human IL-6 ELISA Kit (ab178013); Human TNF alpha ELISA Kit (ab181421); Human IFN gamma High Sensitivity ELISA Kit (ab236895); Human TGF beta 1 ELISA Kit (ab100647) Human IL-17a ELISA kit (ab216167) for IL-1β.

### 4.5. Statistical Analysis

Statistical analysis was performed with Statistica 8, StaSoft, Inc. (Madrid, Spain), and the results are expressed as means ± S.E. All data are expressed as means ± SE and were obtained by one-way ANOVA, and in the LSD post hoc test, *p* > 0.05 was considered statistically significant. To define which groups were different from each other, LSD post hoc tests were used.

## 5. Conclusions

Currently, common biomarkers of lipid peroxidation, such as MDA and 4-HNE, are routinely used to detect and assess the effects of diabetes. It should be noted that the increase in these biomarkers is associated with depletion of intracellular antioxidants in patients with type 2 diabetes mellitus. However, according to our research, studies of other biomarkers are also needed to predict retinopathy in type 2 diabetes mellitus and its most severe complication, diabetic macular edema (DME). In summary, the results of our analysis show a significant overproduction of the following studied oxidative markers in patients with DME—MDA, 4-HNE, NO, IL-17A, TNF-α, and IFN-γ. The EPR method allows for direct measurement of NO in clinical settings as a routine blood test for the prediction of complications of diabetes mellitus.

## Figures and Tables

**Figure 1 ijms-26-03810-f001:**
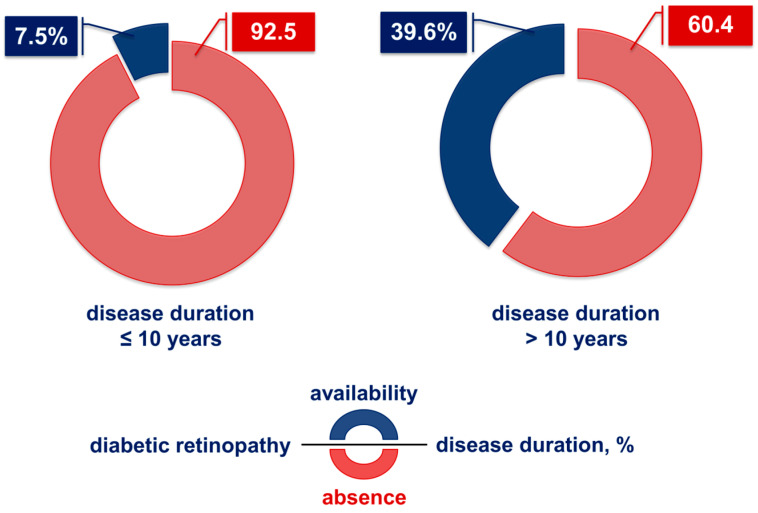
Percentage of diabetic retinopathy according to disease duration (≤10 and >10 years).

**Figure 2 ijms-26-03810-f002:**
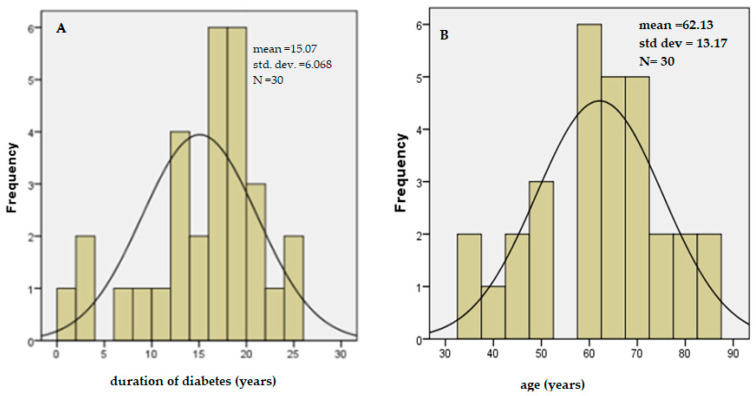
Histogram of the distribution of mean disease duration (**A**) and age (**B**) in patients with DME.

**Figure 3 ijms-26-03810-f003:**
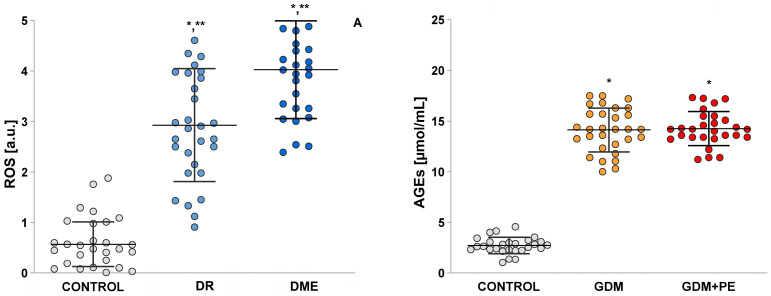
Levels of ROS and advanced glycation end-product AGEs (controls; diabetic retinopathy group, DR; diabetic macular edema group, DME. Least Significant Difference (LSD) post hoc test; (*) *p* < 0.05 vs. control; (**) *p* < 0.05 vs. DME.

**Figure 4 ijms-26-03810-f004:**
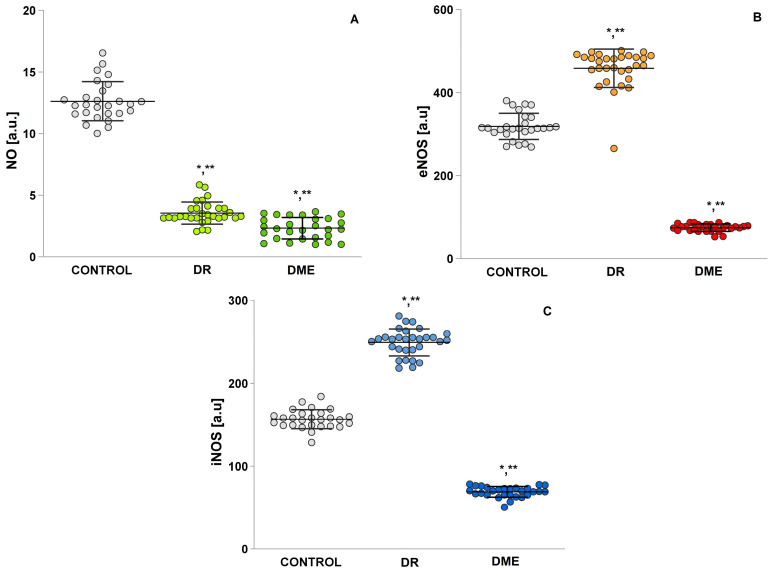
Presented are the levels of nitric oxide (NO), eNOS, and iNOS in serum samples. (**A**) NO: (1) healthy controls; (2) group with diabetic retinopathy, DR; (3) group with diabetic macular edema, DME. (**B**) eNOS: (1) healthy controls; (2) group with diabetic retinopathy, DR; (3) group with diabetic macular edema, DME. (**C**) iNOS: (1) healthy controls; (2) group with diabetic retinopathy, DR; (3) group with diabetic macular edema, DME. LSD post hoc test; (*) *p* < 0.05 vs. control; (**) *p* < 0.05 vs. DME.

**Figure 5 ijms-26-03810-f005:**
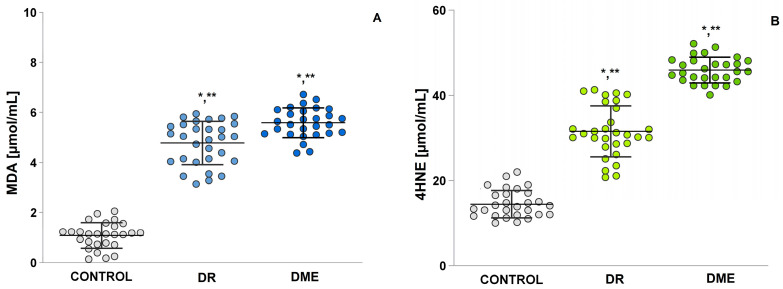
Mean serum levels of MDA (**A**) and 4-HNE (**B**) in the study groups: control group; diabetic retinopathy group, DR; diabetic macular edema group, DME. LSD post hoc test; (*) *p* < 0.05 vs. control; (**) *p* < 0.05 vs. DME.

**Figure 6 ijms-26-03810-f006:**
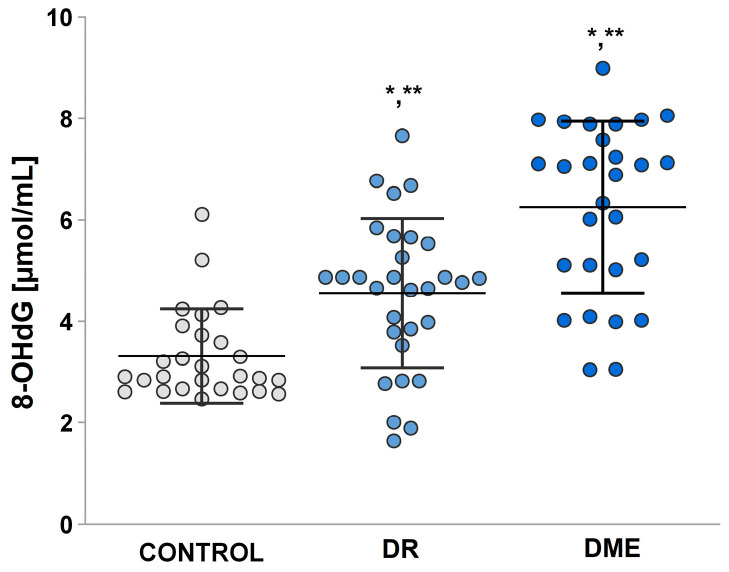
8-OHdG in the studied groups: control group, group with DR, and DME. LSD post hoc test; (*) *p* < 0.05 vs. control; (**) *p* < 0.05 vs. DME.

**Figure 7 ijms-26-03810-f007:**
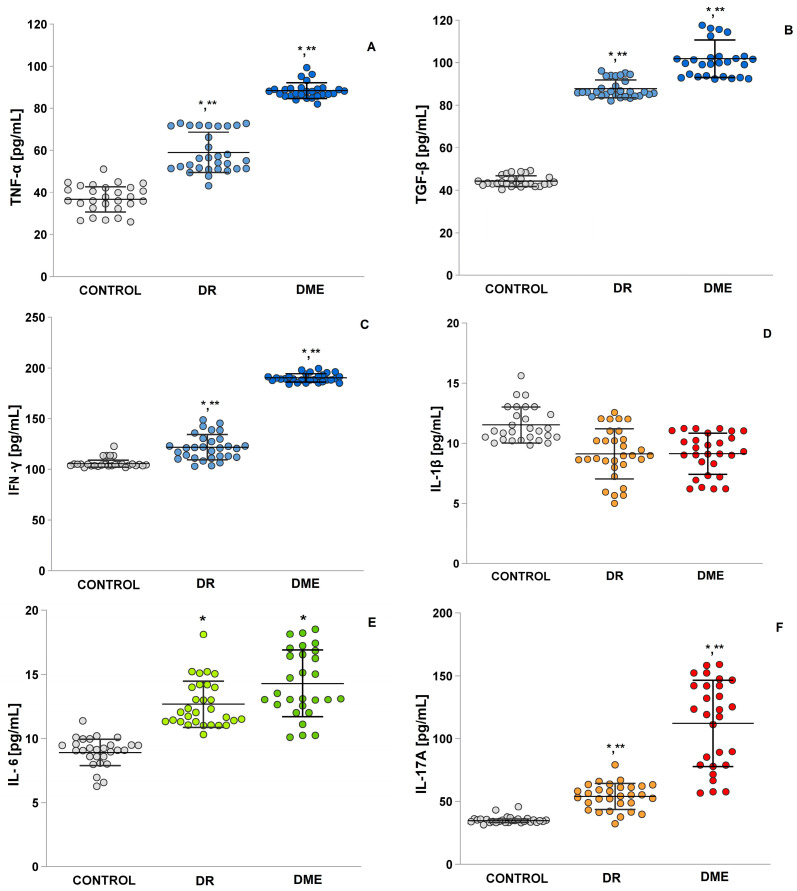
Levels of the markers of oxidative stress, inflammation, TNF-α (**A**), TGF-β (**B**), IFN-γ (**C**), IL-1β (**D**), IL-6 (**E**), and IL-17A (**F**) in the control group, groups with DR, and DME. LSD post hoc test; (*) *p* < 0.05 vs. control; (**) *p* < 0.05 vs. DME.

**Table 1 ijms-26-03810-t001:** Main characteristics of patients with type 2 diabetes mellitus (DMT2) with complications compared to healthy controls. In the control group, there were 94 volunteers, of whom 40.4% were classified as overweight, having a body mass index (BMI) greater than 30 kg/m, n (%). In the diabetic group with retinopathy, 56.2% of the patients were considered obese.

Variables	Controls (n = 94)	DMT2 with DR (n = 96)	DMT2 with DME (n = 38)	*p*
Age; mean ± SD	63.35 ± 12.897	63.15 ± 10.23	62.13 ± 13.177	<0.001
Sex (M/F)	15M/12F	43M/53F	19M/19F	0.948
Disease duration; mean ± SD	-	12.70 ± 8.65	15.07 ± 1.11	-
BMI (kg/m^2^); mean ± SD BMI > 30 kg/m^2^, n (%)	29.23 ± 4.11 38 (40.4%)	31.52 ± 5.89 54 (56.2%)	32.34 ± 6.01 54 (59.01%)	<0.001 <0.001
Blood sugar (mmol/L); mean ± SD	4.97 ± 0.32	9.52 ± 5.18	8.77 ± 0.75	<0.001
HbA1c (%); mean ± SD	5.06 ± 0.27	8.20 ± 2.06	8.21 ± 0.18	<0.001
Cholesterol (mmol/L)	4.43 ± 0.76	5.12 ± 1.38	5.47 ± 0.6	<0.001
Triglycerides (mmol/L)	1.52 ± 0.44	2.43 ± 1.27	2.513 ± 0.17	<0.001
HDL (mmol/L)	1.01 ± 0.28	1.25 ± 0.38	1.95 ± 0.13	<0.001
LDL (mmol/L)	2.32 ± 0.62	2.86 ± 1.10	2.32 ± 0.15	<0.001
SOD U/gHb	121 ± 15.55	58.24 ± 15.31	47.26 ± 13.14	<0.001
CAT U/gHb	48.59 ± 8.66	73.35 ± 8.64	70.87 ± 11.63	<0.001
GPx U/gHb	289.87 ± 25.58	116.49 ± 18.11	76.53 ± 21.38	<0.001

**Table 2 ijms-26-03810-t002:** Comparative analysis of retinal thickness in patients with DR, DME, and control groups.

	Controls	DR	DME
Central macular thickness CMT (μm)	115.62 ± 0.21	226.92 ± 0.844	488.53 ± 0.12

## Data Availability

The data presented in this study are available on request from the corresponding author.

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
