# Peer review of "Systematic Inflammation and Oxidative Stress Elevation in Diabetic Retinopathy and Diabetic Patients with Macular Edema"

_ijms, 2025, doi:10.3390/ijms26083810_

Round 1
Reviewer 1 Report
Comments and Suggestions for Authors
The authors of Manuscript ID: ijms-3481287 entitled, Systematic Inflammation and Oxidative Stress Elevation in Proliferative Diabetic Retinopathy (DR) and Diabetic Patients with Macular Edema (DME) have submitted their original research for peer review as possible for publication in the International Journal of Molecular Sciences.
The manuscript is well prepared and valuable for medical teams.
This study investigated the association between diabetic retinopathy (DR) and its complication, diabetic macular edema (DME), and compared it with biomarkers of oxidative stress. The study aims to compare the main indicators of the development of diabetic retinopathy measured as parameters of oxidative stress and compared to lipid oxidation, DNA damage, and cytokine levels and to monitor their quantitative manifestation in DME. The study evaluated 134 patients, aged 62.10 ± 11.22 years and divided them into two groups: type 2 DM with retinopathy and type 2 DM with macular edema. The authors found increased levels of ROS in the two study groups (DR and DEM). The results of binary logistic regression for the independent influence of ROS showed that they are a risk factor for the development of diabetic retinopathy. Moreover, in patients with T2DM and DME the authors found had significantly higher levels of cytokine production, but decreased NO levels compared to the control group. The study highlights compromised oxidative status as a contributing factor to diabetic macular edema in patients with decompensate type 2 diabetes mellitus. Assessment of oxidative stress levels and inflammatory biomarkers may aid in the early detection and prediction of diabetic complications.
Materials and methods correctly described. In the results section, data on patients and biochemical serum parameters are presented in the tables and figures.
The discussion is conducted honestly using the latest literature.
Author Response
Thank you very much!
Reviewer 2 Report
Comments and Suggestions for Authors
The paper “Systematic Inflammation and Oxidative Stress Elevation in 2 Proliferative Diabetic Retinopathy (DR) and Diabetic Patients 3 with Macular Edema (DME) “ by Petkova-Parlapanska et al., analyzed the markers of inflammation and oxidative stress in diabetic patients tin with ocular compliances.
The paper is potentially interesting but is poorly written and confusing and needs major revision and modification in the study design and subjects.
The authors make a lot of confusion with acronyms making it difficult to read.
In the TITLE Proliferative Diabetic Retinopathy (DR) and Diabetic Patients 3 with Macular Edema (DME); in the ABSTRACT type 2 diabetes 24 mellitus with retinopathy (DMT2 RP) and type 2 diabetes mellitus with macular edema (DMT2 DME); in the FIGURES diabetic retinopathy group T2DMDR, diabetic macular edema group T2DME; in the table and text, random acronyms make everything nebulous…
I ask the Authors to use ONLY TWO ACRONYMS FOR THE WHOLE PAPER.
TITLE - Please remove the 2 acromyms and insert them into the text the first time you mention.
MATERIALS AND METHODS- Study design and subjects and TABLE 1
The authors use 27 slightly overweight people (BMI 25.2) as controls, while the 2 classes of patients, DMT2 with RP and DMT2 with DME (another variant of the acronyms), are people who are classified as obese (BMI > 29.9).
It is known that the levels of inflammatory cytokines and oxidative stress are higher in obese people, so the authors must compare the two groups analyzed to a control group of the same weight class to support their higher levels of inflammation and oxidative stress markers in two class of patients having DT2 and retinopathy.
This is the biggest problem of this paper!
RESULTS - Figure 1 is blurry and absolutely needs to be redone.
Table 3 in this section is out of place; it is not even cited in the results but in a discussion where it should be inserted. The authors must indicate how they calculated it; change EXP B to OR.
Check in the whole text the exact figure cited with the relative panel example Fig 7 missing panel E.
DISCUSSION- I advise the authors to lighten the discussion by inserting the agreed acronyms, in this form is very confusing.
Minor Points
lane 277 NFKB (X) insert the reference
Lane 335 table ?
First the Conclusions and then Materials and Methods
Check the references; example 1, 4, 20
Comments on the Quality of English LanguageThe English could be improved to more clearly express the research.
Author Response
RESPONSES TO THE Reviewers' COMMENTS
We appreciate reviewers’ comments. All corrections in the manuscript are in red.
The paper “Systematic Inflammation and Oxidative Stress Elevation in 2 Proliferative Diabetic Retinopathy (DR) and Diabetic Patients 3 with Macular Edema (DME) “ by Petkova-Parlapanska et al., analyzed the markers of inflammation and oxidative stress in diabetic patients tin with ocular compliances.
The paper is potentially interesting but is poorly written and confusing and needs major revision and modification in the study design and subjects.
The authors make a lot of confusion with acronyms making it difficult to read.
Point 1: In the TITLE Proliferative Diabetic Retinopathy (DR) and Diabetic Patients 3 with Macular Edema (DME); in the ABSTRACT type 2 diabetes 24 mellitus with retinopathy (DMT2 RP) and type 2 diabetes mellitus with macular edema (DMT2 DME); in the FIGURES diabetic retinopathy group T2DMDR, diabetic macular edema group T2DME; in the table and text, random acronyms make everything nebulous…I ask the Authors to use ONLY TWO ACRONYMS FOR THE WHOLE PAPER.TITLE - Please remove the 2 acromyms and insert them into the text the first time you mention.
Answer 1: Thank you very much for help us to improve our manuscript. Everything is done
Point 2: MATERIALS AND METHODS- Study design and subjects and TABLE 1 The authors use 27 slightly overweight people (BMI 25.2) as controls, while the 2 classes of patients, DMT2 with RP and DMT2 with DME (another variant of the acronyms), are people who are classified as obese (BMI > 29.9). It is known that the levels of inflammatory cytokines and oxidative stress are higher in obese people, so the authors must compare the two groups analyzed to a control group of the same weight class to support their higher levels of inflammation and oxidative stress markers in two classes of patients having DT2 and retinopathy. This is the biggest problem of this paper!
Answer 2: We added controls, with the selected volunteers having a higher body mass index. The total control group now consists of 94 people, with 40.4% of the volunteers being overweight with a BMI > 30 kg/m², n (%).
Point 3: RESULTS - Figure 1 is blurry and absolutely needs to be redone.
Answer 3: done
Point 4: Table 3 in this section is out of place; it is not even cited in the results but in a discussion where it should be inserted. The authors must indicate how they calculated it; change EXP B to OR.
Answer 4: the table is deleted
Point 5: Check in the whole text the exact figure cited with the relative panel example Fig 7 missing panel E.
Answer 5: Done
Point 6: DISCUSSION- I advise the authors to lighten the discussion by inserting the agreed acronyms, in this form is very confusing.
Answer 6: Done
Point 7: Minor Points
lane 277 NFKB (X) insert the reference
Lane 335 table?
Answer 7: Done
Point 8: First the Conclusions and then Materials and Methods
Answer 8: We have used the IJMS template
Point 9: Check the references; example 1, 4, 20
RESPONSES TO THE Reviewers' COMMENTS
We appreciate reviewers’ comments. All corrections in the manuscript are in red.
The paper “Systematic Inflammation and Oxidative Stress Elevation in 2 Proliferative Diabetic Retinopathy (DR) and Diabetic Patients 3 with Macular Edema (DME) “ by Petkova-Parlapanska et al., analyzed the markers of inflammation and oxidative stress in diabetic patients tin with ocular compliances.
The paper is potentially interesting but is poorly written and confusing and needs major revision and modification in the study design and subjects.
The authors make a lot of confusion with acronyms making it difficult to read.
Point 1: In the TITLE Proliferative Diabetic Retinopathy (DR) and Diabetic Patients 3 with Macular Edema (DME); in the ABSTRACT type 2 diabetes 24 mellitus with retinopathy (DMT2 RP) and type 2 diabetes mellitus with macular edema (DMT2 DME); in the FIGURES diabetic retinopathy group T2DMDR, diabetic macular edema group T2DME; in the table and text, random acronyms make everything nebulous…I ask the Authors to use ONLY TWO ACRONYMS FOR THE WHOLE PAPER.TITLE - Please remove the 2 acromyms and insert them into the text the first time you mention.
Answer 1: Thank you very much for help us to improve our manuscript. Everything is done
Point 2: MATERIALS AND METHODS- Study design and subjects and TABLE 1 The authors use 27 slightly overweight people (BMI 25.2) as controls, while the 2 classes of patients, DMT2 with RP and DMT2 with DME (another variant of the acronyms), are people who are classified as obese (BMI > 29.9). It is known that the levels of inflammatory cytokines and oxidative stress are higher in obese people, so the authors must compare the two groups analyzed to a control group of the same weight class to support their higher levels of inflammation and oxidative stress markers in two classes of patients having DT2 and retinopathy. This is the biggest problem of this paper!
Answer 2: We added controls, with the selected volunteers having a higher body mass index. The total control group now consists of 94 people, with 40.4% of the volunteers being overweight with a BMI > 30 kg/m², n (%).
Point 3: RESULTS - Figure 1 is blurry and absolutely needs to be redone.
Answer 3: done
Point 4: Table 3 in this section is out of place; it is not even cited in the results but in a discussion where it should be inserted. The authors must indicate how they calculated it; change EXP B to OR.
Answer 4: the table is deleted
Point 5: Check in the whole text the exact figure cited with the relative panel example Fig 7 missing panel E.
Answer 5: Done
Point 6: DISCUSSION- I advise the authors to lighten the discussion by inserting the agreed acronyms, in this form is very confusing.
Answer 6: Done
Point 7: Minor Points
lane 277 NFKB (X) insert the reference
Lane 335 table?
Answer 7: Done
Point 8: First the Conclusions and then Materials and Methods
Answer 9: We have used the IJMS template
Point 9: Check the references; example 1, 4, 20
Answer 9: Done
Reviewer 3 Report
Comments and Suggestions for Authors
1-Novelty is missing in your intro section.
2-Why did the Authors didn’t measure any anti-oxidant parameters?
3-Elaborate the discussion section and improve the coherence with already published similar articles.
4-Cite some relevant :
https://doi.org/10.3390/biomedicines12030552, https://doi.org/10.1016/j.biopha.2023.114772, https://doi.org/10.3390/biomedicines11123202
5-Complete editorial checking will be needed for your manuscript
Comments on the Quality of English LanguageComplete editorial checking will be needed for your manuscript
Author Response
RESPONSES TO THE Reviewers' COMMENTS
We appreciate reviewers’ comments. All corrections in the manuscript are highlighted.
Point 1: Novelty is missing in your intro section.
Answer 1: done
Point 2: -Why did the Authors didn’t measure any anti-oxidant parameters?
Answer 2: done
Point 3: Elaborate the discussion section and improve the coherence with already published similar articles.
Answer 3: done
Point 4: Cite some relevant https://doi.org/10.3390/biomedicines12030552,
https://doi.org/10.1016/j.biopha.2023.114772,
https://doi.org/10.3390/biomedicines11123202
Answer 4: done
Point 5: Complete editorial checking will be needed for your manuscript
Answer 5: done
Reviewer 4 Report
Comments and Suggestions for Authors
This is an interesting paper showing that in diabetic RP and DME had increase in inflammatory cytokines, oxidative lipid product. They also found that eNOS is central in the DME retinopathy. A overview as why you use this approach would be helpful for the reader and then walk us through your experimental plans. What was missing is the significant difference in the HgA1C and the BMI in the T2DM group when compared to controls. The controls have a normal BMI, while all of the DM patients had a BMI in the obese range. It is not clear whether obesity is driving the oxidative stress versus that of uncontrolled sugars. This will significantly change the impact of your finding.
Figure 1 is hard to understand, and thus a better description would help the reader appreciate what you are trying to articulate.
Comments on the Quality of English Language
This was a difficult to understand manuscript despite reading it over a number of times. A schematic as how the authors believe is the foundation of the paper would help this. More granularity as to why these studies were undertaken would really improve the paper.
Author Response
RESPONSES TO THE Reviewers' COMMENTS
We appreciate reviewers’ comments. All corrections in the manuscript are in red.
This is an interesting paper showing that in diabetic RP and DME had increase in inflammatory cytokines, oxidative lipid product. They also found that eNOS is central in the DME retinopathy. A overview as why you use this approach would be helpful for the reader and then walk us through your experimental plans. What was missing is the significant difference in the HgA1C and the BMI in the T2DM group when compared to controls. The controls have a normal BMI, while all of the DM patients had a BMI in the obese range. It is not clear whether obesity is driving the oxidative stress versus that of uncontrolled sugars. This will significantly change the impact of your finding.
Point 1: Figure 1 is hard to understand, and thus a better description would help the reader appreciate what you are trying to articulate.
Answer 1: We have made improvements to Figure 1.
Thank you very much for helping us enhances our manuscript. We hope that the manuscript is now clearer and more detailed. We have added controls, selecting volunteers with a higher body mass index. The total control group now consists of 94 individuals, with 40.4% of the volunteers classified as overweight (BMI > 30 kg/m²).
Round 2
Reviewer 2 Report
Comments and Suggestions for Authors
The paper is accepted in the correct form
Author Response
Thank you!